# Character Beyond Speech: Leveraging Role-Playing Evaluation in Audio Large Language Models via Reinforcement Learning

## Abstract

The advancement of multimodal large model technology has propelled the simulation of diverse characters in speech dialogue systems, establishing a novel interactive paradigm. Character attributes are manifested not only in textual responses but also through vocal features, with speech containing non-semantic information that is challenging to quantify. This poses significant difficulties in evaluating the character embodiment capabilities of role-playing agents. In response to these issues, we present the RoleJudge evaluation framework, which leverages audio large language models to systematically assess the alignment between speech and character across multiple modalities and dimensions. Furthermore, we introduce RoleChat, the first role-playing speech evaluation dataset, comprising both authentic speech samples and detailed reasoning annotations for evaluation. Utilizing this dataset, we implement a multi-stage training paradigm and incorporate standard alignment in reinforcement learning to mitigate reward misalignment during the optimization process. Experimental results on both accuracy and subjective assessment demonstrate that RoleJudge outperforms various baseline models, thereby validating the effectiveness of our multidimensional evaluation framework.

## 1 Introduction

The continuous advancement of artificial intelligence is profoundly transforming the way humans interact with digital systems, giving rise to new forms of digital life that seamlessly integrate technology with human experience. Among these innovations, Role-Playing Agents (RPAs) are particularly noteworthy, as they embody our aspiration to create virtual entities capable of understanding, responding, and interacting with users in increasingly human-like ways. By simulating a wide range of characters, from historical figures and fictional personalities to everyday individuals, these agents open up new possibilities for virtual assistants, interactive storytelling, and intelligent game characters.

Driven by large language models (Bai et al., 2023; Dubey et al., 2024; Fang et al., 2025; Yang et al., 2025), text-based RPAs are gradually becoming a reality (Shanahan et al., 2023; Shao et al., 2023; Wang et al., 2023a), extending to novel applications such as digital humans and character-driven video games (Xu et al., 2024). With the increasing integration of multimodal technologies and large-scale models (SpeechTeam, 2024; Chen et al., 2025b; Zhang et al., 2024a), a subset of RPAs has begun to prioritize direct human-computer interaction through voice-based communication. (Zhang et al., 2025) Beyond semantic content, spoken language conveys paralinguistic cues, including style and emotion, that are fundamental to the expression of the character's personality. Achieving optimal alignment between model-generated outputs and predefined character profiles necessitates the production of voice dialogues that faithfully emulate the intended character, thereby enhancing user immersion. Consequently, a critical challenge has emerged: assessing whether the speech generated by RPAs authentically embodies the character and systematically exploring character traits that extend beyond the surface-level linguistic content.

The evaluation of textual outputs generated by RPAs constitutes a vibrant area of research, where authentic character dialogue data sourced from films, novels, and games are utilized to assess agents

Figure 1: RoleChat encompasses five evaluation dimensions: Logical Coherence, which assesses the logical soundness of the response text; Content Relevance, which evaluates whether the response aligns with the character information; Context Consistency, which measures the semantic coherence across multiple dialogue turns as well as the smoothness of emotional transitions; Emotional Appropriateness, which examines the plausibility of the expressed emotions in the response; and Style Alignment, which determines whether the vocal style matches the character.

across dimensions such as interaction capability, character consistency, and user engagement (Tu et al., 2024; Chen et al., 2024a; Feng et al., 2025). In contrast, spoken language introduces complex acoustic information that is absent in textual modalities, and the nuanced interplay between these acoustic features and character traits renders the evaluation of voice-based RPAs both highly subjective and methodologically challenging. As a result, conventional text-based benchmarks are insufficient for assessing spoken outputs, leaving the evaluation of voice-enabled RPAs as an open research problem. Nonetheless, recent advancements in audio foundation models present promising avenues and novel methodologies for addressing these challenges. Audio foundation models are developed to address a broad spectrum of challenges within the audio and speech domains (Tang et al., 2023; Xu et al., 2025; Chen et al., 2024b; Ghosh et al., 2025; Zhang et al., 2024b). Nevertheless, supervised fine-tuning (SFT) on task-specific datasets often constrains their evaluative capabilities, as these models are predominantly optimized for generation or recognition tasks rather than assessment. Recent efforts have sought to enhance the evaluation capacity of audio models by constructing paired speech-evaluation datasets, targeting applications such as synthetic audio quality assessment (Chen et al., 2025a) and the evaluation of intelligence and emotional quotient in spoken dialogues (Ji et al., 2025). Despite these advancements, the application of such methodologies to the evaluation of RPAs presents two primary challenges: (1) Existing evaluation approaches are typically uni-dimensional, yielding a single score that fails to encapsulate the multifaceted nature of speech quality; (2) The SFT paradigm inherently limits model generalization, which is essential for handling diverse evaluative tasks. Furthermore, reinforcement learning-based methods are highly sensitive to data quality. When reward signals are sparse, models are prone to deviating from the global optimum and getting trapped in local optima due to insufficient feedback (Guo et al., 2025), which ultimately impairs overall performance.

Io the light of these challenges, we introduce RoleChat, the first evaluation dataset for role-playing dialogue, comprising 50 distinct characters and 14,032 samples. The dataset consists of both collected and large model-generated samples, with each sample containing character information, dialogue history, user queries, and model outputs. For identical dialogue histories, we sample di-

verse model outputs to enable a more comprehensive understanding of conversations from multiple perspectives. Each sample is annotated with detailed reasoning and scored across five evaluation dimensions: Logical Coherence, Content Relevance, Context Consistency, Emotional Appropriateness, and Style Alignment, as illustrated in Figure 1. The quality of both the speech data and evaluation scores is rigorously ensured. Building upon this dataset, we propose a multidimensional evaluation framework, RoleJudge. Expert models are trained on different evaluation dimensions, and a subset of RoleChat data is utilized for supervised fine-tuning of audio large models to achieve cold-start initialization, equipping the models with fundamental task comprehension and appropriate output formatting capabilities. Subsequently, we employ standard alignment reinforcement learning, where, based on the GRPO framework (Guo et al., 2025), authentic or high-scoring samples are introduced as standards. The model's understanding of these standard samples represents its evaluative performance on corresponding tasks. The average reward of standard samples is used as a scaling parameter for other samples with identical query, preventing the model from selecting relatively high-reward actions in scenarios with low absolute rewards and thus avoiding local optima. Finally, the weights of expert models are integrated to obtain the final model. Our main contributions are as follows:

- RoleJudge is the first evaluation model specifically designed for voice-based role-playing dialogue. It takes speech-to-speech conversations as input and assesses the quality of responses from multiple perspectives, including text and speech multimodality, as well as alignment and consistency. Extensive experiments demonstrate the effectiveness of RoleJudge.
- RoleJudge introduces standard rewards as absolute guidance in positive and negative multi-sample sampling, optimizing the alignment of reward signals under relative reward settings and thereby enhancing the model's evaluative capacity.
- We present RoleChat, the first role-playing dialogue evaluation dataset comprising dialogue-score pairs. RoleChat not only includes a diverse set of speech responses generated by large models, but also samples a portion of authentic data as high-quality reference responses.

## 2 RELATED WORKS

### 2.1 ROLE-PLAYING AGENTS.

Role-Playing Agents (RPAs) are intelligent agents capable of simulating the knowledge, behaviors, emotions, and communication styles of specific characters, thereby achieving highly anthropomorphic role-playing abilities (Shanahan et al., 2023; Shao et al., 2023). RPAs typically leverage capabilities such as in-context learning, instruction following, and social intelligence to reproduce the linguistic and behavioral characteristics of historical figures, fictional characters, or real individuals (Zhou et al., 2024). The outstanding performance of large language models (LLMs) in generating human-like content has greatly propelled the development of RPAs. Some works employ retrieval-augmented generation (RAG) and similar methods to enable agents to reproduce character-specific knowledge (Li et al., 2023), while other studies focus on aligning the linguistic style with the target persona (Wang et al., 2023b), and yet others aim to train agents with profile and experience perception to reflect deeper personality traits (Lu et al., 2024). Recently, with the advancement of multimodal technologies, RPAs have gradually expanded to include multimodal features such as voice style. For example, OmniCharacter seamlessly integrates speech and language to ensure immersive interactions for RPAs Zhang et al. (2025).

As the application scope of RPAs continues to expand, the evaluation of LLMs' role-playing abilities has become increasingly important. RoleEval (Shen et al., 2023) constructs a bilingual benchmark dataset and designs multiple-choice questions to assess models' capabilities in character knowledge acquisition, understanding, and reasoning. On the other hand, TimeChara (Ahn et al., 2024) focus on evaluating the ability of models and agents to identify and correct errors. CharacterEval (Tu et al., 2024) introduces multiple evaluation metrics and establishes a scoring standard for assessing the role-playing effectiveness of models. To facilitate the evaluation of subjective indicators, a human-annotated role-playing reward model, CharacterRM, has been developed. However, text-based dialogue evaluation methods are not well-suited for role-playing voice dialogue scenarios, which are more common and direct in practical applications. Therefore, a more comprehensive assessment framework is required for evaluating RPAs.

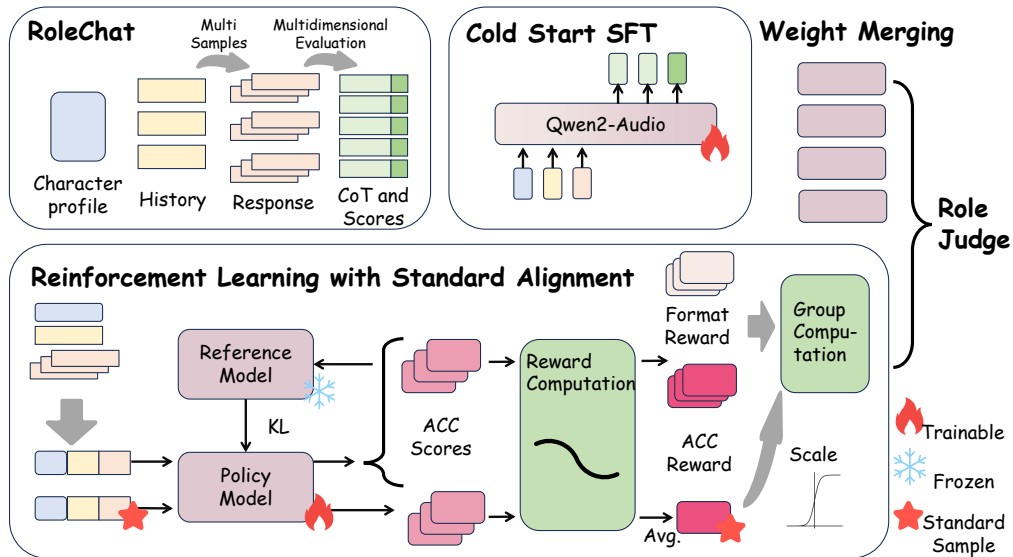

Figure 2: The overall architecture of RoleJudge. It comprises initial model supervised fine-tuning and standard alignment reinforcement learning with single-dimension, followed by weight merging to construct a unified multidimensional evaluation model. Leveraging an audio large model backbone, RoleJudge facilitates joint understanding of textual and acoustic modalities, thereby enabling fine-grained analysis and holistic assessment of role-playing dialogues.

## 2.2 LLMs FOR SPEECH INFORMATION PERCEPTION.

In recent years, the development of multimodal technologies has enabled the alignment of audio modalities with large model inputs, thereby facilitating extensive audio understanding by large language models. Some studies encode speech into discrete tokens and incorporate them into LLMs, allowing the models to accept audio input, as seen in works such as SpeechGPT (Zhang et al., 2023) and AudioPaLM (Kong et al., 2024). Models like SALMONN (Tang et al., 2023) and Qwen-Audio (Chu et al., 2023; 2024) are trained on large-scale, multi-task datasets, equipping them to perform a variety of downstream tasks including speech recognition, speech translation, and audio event detection. A subset of research applies large audio models to spoken dialogue, enabling more intelligent interactions, for example, by mining paralinguistic factors such as style to generate emotionally rich responses (Lin et al., 2024), or by avoiding cascaded approaches to achieve more real-time interaction. (Zeng et al., 2024)

Recently, some studies have explored the potential of large audio models in evaluating speech-related tasks. For instance, reinforcement learning has been introduced for the first time, utilizing large audio models as descriptive speech quality evaluators to assess TTS outputs and achieve more accurate evaluation (Chen et al., 2025a). WavReward (Ji et al., 2025) further extends this approach by employing chain-of-thought reasoning, using models to evaluate both the intelligence and emotional quotient of end-to-end spoken dialogue systems. These works demonstrate the enhanced generalization capabilities of reinforcement learning in evaluation tasks. However, when facing the multidimensional requirements of role-playing evaluation, the training strategies still require redesign, and high-quality datasets are essential, as annotation errors can adversely affect the learning of reward signals.

To address these challenges, we have constructed RoleChat, a dataset specifically designed for role-playing dialogue evaluation, encompassing five dimensions of assessment. We introduce reinforcement learning with standard alignment, introducing model performance as an absolute score to scale the advantages within positive and negative sample groups, thereby reducing the occurrence of selecting the best among suboptimal options. This approach effectively improves the accuracy of models in role-playing evaluation tasks.

# 3 ROLE JUDGE: MULTIDIMENSIONAL EVALUATION FRAMEWORK

## 3.1 OVERVIEW

Following the training framework of DeepSeek-R1 (Guo et al., 2025), the overall pipeline of Role Judge consists of supervised fine-tuning (SFT) with a subset of data for cold-start initialization, subsequent reinforcement learning-based post-training with standard alignment, and expert model weight, as illustrated in Figure 2. The baseline model for Role Judge is Qwen2-Audio Chu et al. (2024), which demonstrates strong performance across various audio-related tasks. On the input side, the large language model leverages the alignment between the audio encoder and the language model, enabling simultaneous comprehension of both semantic and acoustic information within speech. Compared to cascaded approaches that separately extract audio features and utilize text-based large language models, Qwen2-Audio is better suited for the evaluation of voice-based role-playing agents.

We define the evaluation task for role-playing speech as follows: Given the character profile $P$, the dialogue history sequence $\{h_0, h_1...h_k\}$ between the role and the user, the current user query $q$, and the agent's response $t$, the evaluation model is required to understand $t$ from both semantic and acoustic perspectives. Integrating all available information, the model must assess the agent's speech output across five dimensions: response rationality, response consistency, historical coherence, emotional appropriateness, and stylistic alignment. The model should output both the chain-of-thought reasoning process $c_i$ and the final scores $s_i$, with $i$ representing evaluations dimensions.

To enhance the model's ability to distinguish between different evaluation tasks and deepen its understanding of speech across multiple dimensions, we train separate expert models for each dimension and subsequently merge their weights to construct a unified evaluation system. For model training, we directly concatenate the encoded representations of textual and audio information as the input, thereby improving the model's capability for multimodal comprehension.

## 3.2 COLD-START SUPERVISED FINE-TUNING

To ensure a stable and effective reinforcement learning trajectory, we initiate the training process with a cold-start supervised fine-tuning (SFT) phase. During this stage, the model is trained on a curated dataset consisting of paired audio-text samples annotated with detailed chain-of-thought reasoning and multidimensional quality scores. The objective of SFT is to equip the model with a foundational understanding of the evaluation task and the required structured output format, thereby providing a robust starting policy for subsequent reinforcement learning.

Formally, given a batch of $N$ training samples $\{(x_i, y_i)\}_{i=1}^{N}$, where $x_i$ denotes the concatenated input and $y_i$ represents the structured output (including reasoning and scores), we minimize the following cross-entropy loss:

$$\mathcal{L}_{\text{SFT}} = -\frac{1}{N} \sum_{i=1}^{N} \log P_\theta(y_i \mid x_i) \tag{1}$$

where $P_\theta(y_i \mid x_i)$ is the probability of the model with parameters $\theta$ generating the target output $y_i$ given the input $x_i$. This supervised fine-tuning stage is crucial for aligning the model's initial behavior with human-annotated standards and structured reasoning, which facilitates efficient exploration and optimization in the subsequent reinforcement learning phase.

## 3.3 REINFORCEMENT LEARNING WITH STANDARD ALIGNMENT

In large-scale model training, reinforcement learning methods are widely used to optimize the quality of generated outputs. Classic algorithms such as Proximal Policy Optimization (PPO) (Schulman et al., 2017; Yu et al., 2022), which relies on a separately defined value function (critic), and Direct Preference Optimization (DPO) (Rafailov et al., 2023), which leverages preference signals between candidate outputs and reference answers, have achieved remarkable success. However, these approaches face challenges in tasks such as role-playing speech evaluation, which require complex acoustic understanding—the value function is difficult to define and preference signals are hard to align.

Group Relative Policy Optimization (GRPO) (Guo et al., 2025) introduces a group-based sampling paradigm that is better suited for modeling differences in speech across various dimensions. For each sample within a specific evaluation dimension, we collect diverse feedback and assign scores. The mean reward within the group serves as the baseline output, and the relative advantage estimation is incorporated into the reward objective.

Considering that the model generates both reasoning processes $c$ and scoring results $s$, we define the reward objective as comprising two parts: $r_a$ and $r_f$, representing the accuracy reward and the format reward, respectively. The format reward enforces strict adherence to the required output structure, if the model's output conforms to the specified tag format, a reward of 1 is assigned; otherwise, 0. This ensures the evaluation model produces consistently structured outputs. For accuracy reward, we adopt a non-linear approach inspired by previous work in speech evaluation tasks (Ji et al., 2025), $R_a(s, s_c) = 10 \cdot \exp\left(-\frac{(s_c - s)^2}{2\sigma^2}\right)$ where $s_c$ denotes the annotated score for the sample. $r_a$ decreases exponentially as the scoring difference increases, encouraging the model to achieve higher accuracy.

The GRPO method estimates relative advantage within a group by normalizing rewards, thereby reducing computational cost. However, when the overall quality of model outputs is low, relying exclusively on group-relative scores may result in the model favoring outputs that are only relatively better within a poor-performing group, rather than genuinely high-quality responses. To mitigate this, we introduce ground-truth data and scores as a standard reward for absolute evaluation. A key characteristic of role-playing evaluation datasets is that they can be mined from real scenarios, meaning most collected data contain standard answers. If the model's reasoning aligns with the standard answer, it demonstrates a thorough understanding of the speech. Thus, by performing group sampling and reward calculation on standard samples, we use the average reward $r_u$ as a scaling factor in the advantage estimation for both positive and negative samples sharing the same query. Specifically, we modify the advantage estimation as follows:

$$A_i = \text{scale}(r_u) \frac{r_i - \frac{1}{N} \sum_{i=1}^{N} r_i}{\sqrt{\frac{1}{N} \sum_{i=1}^{N} \left(r_i - \frac{1}{N} \sum_{i=1}^{N} r_i\right)^2}} \tag{2}$$

where $\text{scale}(r_u) = a + (b - a) \cdot \text{sigmoid}(\alpha(r_u - 0.5))$. This represents a smooth scale alignment based on the standard reward, where $a$ and $b$ control the minimum and maximum scaling factors, and $\alpha$ determines the sharpness of the transition. In essence, if the model performs poorly on standard samples, it indicates insufficient evaluative capability for the given query. Even if the relative advantage is large, it may not represent an optimal direction for model improvement and could lead to local optima. Therefore, by scaling the advantage, we reduce the magnitude of model updates for that query, thereby mitigating the risk of suboptimal convergence.

By introducing the standard reward, the model achieves greater advantage when its overall performance is good, leading to more accurate and higher-quality evaluations. To balance the low reward and vanishing gradients caused by uniformly poor outputs, we use the standard reward as a difficulty coefficient to weight the total reward. If the standard reward is large (indicating poor model performance), the reward function emphasizes format-based rewards:

$$R = \pi r_u \cdot r_a + (1 - \pi r_u) \cdot r_f \tag{3}$$

where $\pi r_u$ is the difficulty coefficient derived from the standard reward $r_u$, and $\pi$ is a scaling hyperparameter. The calculation of KL divergence and the reinforcement learning loss function are kept consistent with GRPO.

### 3.4 WEIGHT MERGING

After training expert models for each evaluation dimension, we integrate them into a unified multi-dimensional role-playing dialogue evaluation model using a weighted parameter merging strategy. Each expert model specializes in a specific aspect of speech quality. We assign weights based on the relevance and frequency of evaluation tasks for each dimension, and perform weighted averaging of model parameters. The resulting merged model combines the fine-grained judgment capabilities of all experts, enabling comprehensive and efficient assessment across multiple dimensions in a single inference. This approach enhances both generalization and evaluation efficiency, providing a robust foundation for practical multidimensional role-playing dialogue evaluation.

# 4 ROLE CHAT: FIRST ROLE PLAYING DIALOGUE EVALUATION DATASET

## 4.1 OVERALL

To enable models to accurately assess the quality of role-playing speech from multiple dimensions, we present role-chat, the first large-scale evaluation dataset encompassing role-playing dialogues. This dataset features comprehensive character profiles and provides diverse responses—including both positive and negative examples—for identical scenarios, as well as a subset of real speech data. Each dialogue sample is annotated with multi-dimensional reasoning and scoring. To ensure the high quality of the dataset, we have established a rigorous and systematic data construction pipeline.

## 4.2 DATASET CONSTRUCTION

**Stage 1: Character Profile Construction.** To collect authentic speech data, we curate 50 virtual characters from films, television dramas, and other audiovisual works. To ensure the uniqueness of each character profile, we conducted a detailed summary of their personal information. We gather background information, key plot points, and selected lines from these works, and leverage the powerful generative capabilities of large language models (OpenAI et al., 2024) to extract and summarize character details, forming comprehensive profiles that include personality traits, experiences, hobbies, and habits. Subsequently, all profiles are manually verified and any unfaithful information was removed to ensure the accuracy of character identities.

**Stage 2: Dialogue Text Generation.** For the generation of textual dialogues, we adopted a dual approach to construct dialogue histories and user queries. One approach involves collecting authentic dialogue histories directly from film and television works, ensuring the data reflects real-world scenarios and remains faithful to the character's persona. The other approach utilizes synthetic historical scenarios, where we employ GPT-4 (OpenAI et al., 2024) to generate plausible interactions between characters and users, covering a wide range of topics such as daily life, character experiences, and personal viewpoints. We explicitly require that character utterances do not contradict their profiles, thereby guaranteeing the accuracy of the dialogue history. For the final character responses, the segments to be evaluated, we use models from the Qwen2.5 series (Bai et al., 2023) of various sizes, as well as the GPT series (OpenAI, 2024; OpenAI et al., 2024), to generate diverse replies, sampling a range of response qualities to enrich the evaluation dataset.

**Stage 3: Dialogue Speech Generation.** During the speech dialogue generation phase, for synthetic historical scenarios, we leverage existing character audio and apply zero-shot TTS with CosyVoice (Du et al., 2024) to construct character speech for the dialogue history. For character responses, we randomly select different audio samples from the same character, from other characters, or use the TTS model's default voice settings with randomly assigned emotions, intonation, speed, and accent to generate a variety of speech samples, thereby maximizing acoustic diversity. Since the reference audio already contains attributes such as emotion and character style, randomly selecting reference samples enables the construction of speech outputs with diverse styles. Additionally, incorporating audio from different characters and instruction-based TTS further enriches the stylistic diversity of the samples. After generating speech samples, we employ the SenseVoice model (SpeechTeam, 2024) for ASR and filter out samples with high WER to ensure the quality of the synthesized speech.

**Stage 4: Data Scoring.** For sample reasoning and scoring, we utilize the state-of-the-art audio understanding model Gemini-2.5 Pro (Comanici et al., 2025) to provide detailed descriptions of the generated audio. Based on these descriptions, we use the GPT-4 to perform reasoning and scoring. To ensure the reliability of reasoning and scoring, multiple volunteers were recruited to manually verify and refine the results. For synthetic historical scenarios, we ensure that for each query, there exists at least one sample with the highest score in each evaluation dimension, which is selected as the standard sample for that dimension.

# 5 EXPERIMENTS

## 5.1 DATASETS AND BASELINES.

We partitioned 10% of the RoleChat dataset as the evaluation set, within which three roles are included that do not appear in the training set, in order to assess the model's generalization capability

Table 1: Accuracy performance of RoleJudge and other baselines on RoleChat across multi evaluation dimensions: Logical Coherence (L-C), Content Relevance (C-R), Context Consistency (C-C), Emotional Appropriateness (E-A), and Style Alignment (S-A), Overall Acc and Format Acc. The bolded scores indicate the best performance achieved in each respective dimension.

| Method | | Textual | | Spoken | | C-C | Overall Acc | Format Acc |
| | | L-C | C-R | E-A | S-A | | | |
|---|---|---|---|---|---|---|---|---|
| **Text-Modality** | *Open-Source Models* | | | | | | | |
| | Qwen3-7B | 78.2 | 75.4 | - | - | - | - | 91.1 |
| | Qwen3-32B | 82.1 | 76.7 | - | - | - | - | 94.3 |
| | *Closed-Source Models* | | | | | | | |
| | GPT-4 | **96.6** | **92.1** | - | - | - | - | **100** |
| **Multi-Modality** | *Open-Source Models* | | | | | | | |
| | SALMONN-7B | 11.2 | 23.2 | 43.2 | 12.1 | 22.1 | 22.36 | 6.2 |
| | Qwen-Audio | 35.2 | 29.3 | 34.2 | 16.2 | 32.3 | 29.48 | 0 |
| | Qwen2-Audio | 40.9 | 25.1 | 42.1 | 11.1 | 34.1 | 30.66 | 10.2 |
| | Qwen2.5-Omni | 62.8 | 43.1 | 51.1 | 21.3 | 35.3 | 42.72 | 73.2 |
| | *Closed-Source Models* | | | | | | | |
| | GPT-4o-audio | 65.2 | 42.3 | 61.2 | 52.4 | 44.2 | 53.06 | 94.2 |
| | Gemini2.5 Pro | 86.2 | 72.7 | 75.8 | 51.2 | 62.1 | 69.60 | **100** |
| | RoleJudge | 95.1 | 90.2 | **85.2** | **76.0** | **84.0** | **86.10** | **100** |

to unseen roles after training. Furthermore, each role's dialogues contain samples entirely drawn from real-world scenarios, enabling us to test whether the model can maintain accurate evaluation performance in authentic settings.

To comprehensively evaluate the role-playing assessment capability of RoleJudge, we compared multiple large model-based approaches across different modalities, model sizes, and architectures. These include single-text modality open-source models such as Qwen3-8B and Qwen3-32B (Yang et al., 2025), as well as the proprietary GPT-4 (OpenAI et al., 2024), which are used to specifically assess role-playing evaluation from a text perspective (with ASR results as input). For multimodal audio models, we included open-source models such as SALMOON, Qwen-Audio (Chu et al., 2023), Qwen2-Audio (Chu et al., 2024), and Qwen2.5-Omni (Xu et al., 2025), as well as proprietary models GPT-4o-Audio (OpenAI, 2024) and Gemini2.5Pro (Comanici et al., 2025).

For evaluation metrics, we adopted accuracy to measure the discrepancy between the predicted scores and the annotated scores. We assessed the model's understanding ability from five dimensions: Logical Coherence, Content Relevance, Context Consistency, Emotional Appropriateness, and Style Alignment. We also calculated the average accuracy to evaluate the model's overall capability, and a format accuracy metric to assess whether the model can follow instructions and generate the correct reasoning and evaluation structure.Furthermore, we invited volunteers to participate in our data construction process, generating dialogue data through real-time interactions and conducting A/B testing of the evaluation models.

## 5.2 MAIN RESULTS

As shown in Table 1, we evaluated the performance of RoleJudge and other baseline models on the RoleChat test set, demonstrating that RoleJudge achieves the best overall evaluation results, surpassing all baseline models across different modalities and model sizes. From the perspective of text-based metrics, RoleJudge performs slightly below its teacher model, GPT-4, which is expected given the significant disparity in model scale and the fact that GPT-4 served as the annotator for the training data. Nevertheless, RoleJudge achieves higher evaluation accuracy than the Qwen3 models of similar size, highlighting its superior semantic understanding capabilities.Compared to other large audio models with similar input modalities, RoleJudge consistently outperforms the baselines across all evaluation dimensions. Notably, in identity-related tasks such as Content Relevance

Table 2: A/B Test result for RoleJudge.

| Models | RoleJudge Win ↑ | RoleJudge Lose ↓ |
|---|---|---|
| Qwen2.5-Omni | 91 | 9 |
| Gemini2.5Pro | 82 | 18 |

and Style Alignment, RoleJudge exceeds the best baseline model, Gemini2.5Pro, by 17.5 and 24.8 points, respectively. These results demonstrate that the RoleChat dataset and reinforcement learning training framework contribute significantly to the improved role-playing evaluation performance of RoleJudge.Furthermore, there remains a considerable gap between text and audio modalities in terms of logical and content relevance capabilities, indicating that there is still substantial room for improvement in the design of multimodal large models.

## 5.3 A/B Test for RoleJudge

A/B testing is a common subjective evaluation method in which human listeners compare two output results and select the one with higher quality. We recruited ten volunteers who, following a process similar to our data construction, interacted with randomly selected models and randomly assigned TTS role-playing agents to generate ten samples each. These samples were then evaluated and scored by RoleJudge, Qwen2.5-Omni, and Gemini2.5Pro. The volunteers performed pairwise comparisons based on the evaluation results and selected the higher-quality option. As shown in Table 2, RoleJudge achieved a significant advantage over the other two models, indicating that its scoring system demonstrates superior performance in real-world scenarios.

## 5.4 Ablation Experiments

We conducted experiments using three different configurations: training a single model on all data without weight merging, applying GRPO without standard alignment, and performing only supervised fine-tuning (SFT), as shown in Table 3. The results indicate that each component contributes to the overall performance improvement of the model, with reinforcement learning in particular yielding a 14.6-point increase, demonstrating its effectiveness in enhancing the model's generalization capability.

Table 3: Ablation experiments for RoleJudge.

| Reforcement Learning | Standard Alignment | Weight Merging | Overall ACC | Format ACC |
|---|---|---|---|---|
| ✔ | ✔ | ✘ | 82.10 | 100 |
| ✔ | ✘ | ✘ | 78.81 | 100 |
| ✘ | ✘ | ✘ | 64.21 | 85.2 |

## 6 Conclusion

In this work, we construct RoleChat, the first dataset for role-playing dialogue evaluation, featuring multidimensional assessment and reasoning annotations. To fully leverage this dataset, we develop a multi-stage training paradigm, including cold-start supervised fine-tuning, reinforcement learning, and weight merging. Furthermore, we incorporated standard alignment into the reinforcement learning process, scaling the advantage for more challenging tasks and alleviating issues related to misaligned reward optimization directions. Evaluations on accuracy and A/B testing demonstrate the effectiveness of both the dataset and the proposed methods, providing a solid foundation for the development of role-playing speech agents.

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

## APPENDIX

## A    USE OF LLM

In this work, we employed large language models (LLMs) for generating the dialogue data of RoleChat.

## B    EXPERIMENTAL SETUP

We implemented the roleplaying MULTIDIMENSIONAL EVALUATION FRAMEWORK based on the Qwen2-Audio-7B-Instruct model. The training process is divided into two stages: **Cold-Start Phase**: This phase aims to enable the model to understand the task and generate reasoning and scores in the correct format. The learning rate is set to $1 \times 10^{-5}$, the batch size is 4, and training is performed on 8 A100 GPUs. **Reinforcement Learning Phase**: In this phase, we expect the model to accurately comprehend and evaluate speech data across different dimensions. We train five expert models independently, with hyperparameters set as a learning rate of $5 \times 10^{-7}$, batch size of 2, scaling hyperparameters $a = 0.5$, $b = 1.5$, $\alpha = 8$, and $\pi = 0.8$, as well as a KL-divergence regularization beta value of $0.01$. Training is performed on 32 A100 GPUs.

## C    DATASET STATISTICS

RoleChat comprises a total of 50 characters and 14,032 samples, corresponding to 140.2 hours of audio data. We allocate 75% of the samples for supervised fine-tuning (SFT), 15% for reinforcement learning, and 10% for testing. Further details regarding the distribution of character speech lengths in the dialogue history, user speech lengths, and the response speech lengths are illustrated in Figure 3 (a), while the word cloud visualization is presented in Figure 3 (b).

702
703
704
705
706
707
708
709
710
711
712
713
714
715
716
717
718
719
720
721
722
723
724
725
726
727
728
729
730
731
732
733
734
735
736
737
738
739
740
741
742
743
744
745
746
747
748
749
750
751
752
753
754
755

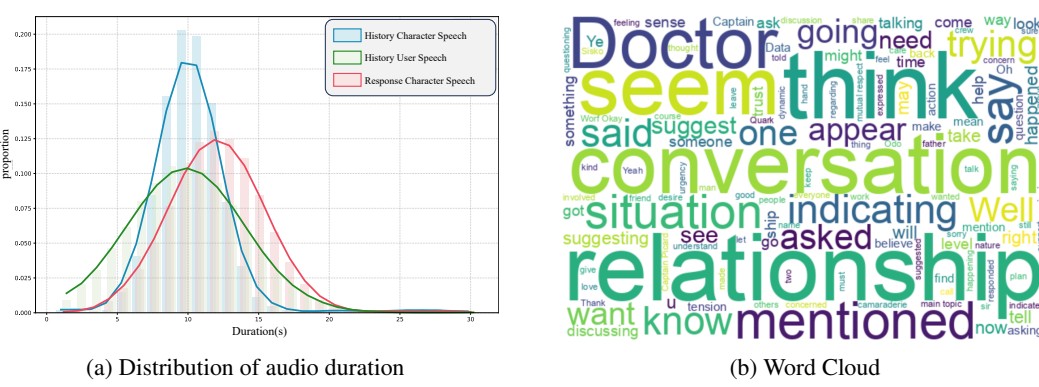

(a) Distribution of audio duration           (b) Word Cloud

Figure 3: Statistics of RoleChat.

