# OpenReview forum: "Character Beyond Speech: Leveraging Role-Playing Evaluation in Large Audio Language Models via Reinforcement Learning"
_ICLR.cc/2026/Conference — ICLR 2026 Conference Withdrawn Submission_

### Official Review · Reviewer_NSat · 2025-10-27

**Soundness:** 1
**Presentation:** 2
**Contribution:** 2
**Rating:** 2
**Confidence:** 4

**Summary:**

This paper presents **RoleJudge**, a novel, multi-dimensional evaluation framework designed to assess the character embodiment capabilities of **speech-enabled Role-Playing Agents (RPAs)**. RoleJudge leverages Audio Large Language Models (Audio LLMs) to systematically evaluate the alignment between a model's speech response and its predefined character profile across five critical dimensions: logic, content, context, emotion, and style.

To support the framework, the authors introduce **RoleChat**, the first large-scale role-playing speech evaluation dataset, which includes both authentic and generated speech samples with detailed, multi-dimensional reasoning annotations. The training methodology utilizes a multi-stage paradigm, notably incorporating **Standard Alignment** into the RL phase. This mechanism uses high-score or real samples as an absolute "standard" to calibrate the reward signal, mitigating reward misalignment and preventing the model from converging to suboptimal local minima.

**Strengths:**

1.  **Pioneering Solution to a Critical Evaluation Gap:** The paper introduces the first multi-dimensional evaluation framework, **RoleJudge**, specifically for **speech-based role-playing dialogues**. This is a significant contribution that fills an important gap in the field by successfully integrating hard-to-quantify vocal non-semantic information (style and emotion) into the evaluation system, moving beyond traditional text-only assessment.
2.  **High-Quality and Comprehensive Dataset Contribution:** The proposed **RoleChat** dataset is a valuable resource, being the first large-scale evaluation dataset for role-playing speech. It covers 50 distinct characters and 14,032 samples. Crucially, it provides detailed **multi-dimensional annotations and CoT reasoning**, which is essential for training robust LLM-based judges.
3.  **Innovative Reinforcement Learning Strategy:** The proposed **Standard Alignment RL** strategy is a novel approach to reward calibration. By using the average reward of high-quality standard samples as a scaling factor, it effectively addresses the known issue of **reward misalignment and local optima** that can occur in relative ranking-based RL frameworks (like GRPO), enhancing the accuracy and stability of the assessment model.
4.  **Clear Modular Design for Multi-Dimensional Assessment:** The design of training separate **Expert Models** for five distinct evaluation dimensions and subsequently merging their weights is methodologically sound. This modularity demonstrates a clear logic for capturing the nuanced differences in speech quality across various attributes, enhancing the comprehensiveness and reliability of the overall evaluation.

**Weaknesses:**

1.  **Incomplete Evaluation Baseline Comparison:** While the paper compares RoleJudge against several LLM-based baselines, it lacks a comparison against strong **unimodal baselines** [1,2,3,4]. Including these comparisons is necessary to empirically validate that the **multimodal integration** of RoleJudge truly yields superior performance over the best single-modality alternatives.
2.  **Objectivity and Bias of LLM-as-a-Judge Paradigm:** The scoring and reasoning process heavily relies on large models like **GPT-4 and Gemini-2.5 Pro**, inheriting the inherent biases and limitations of the **LLM-as-a-Judge** paradigm. The paper should provide a more in-depth **Human-AI Agreement study** (e.g., Cohen's Kappa score or inter-rater reliability) and discuss the limitations of using LLMs to evaluate subjective attributes like **vocal style** and **emotional authenticity**.
3.  **Sensitivity of Reinforcement Learning Hyperparameters:** The Standard Alignment RL introduces several key hyperparameters ($\alpha$, $\pi$, $\sigma^2$, etc.) that are critical for model convergence and final performance. Although values are provided in the appendix, the paper **lacks a sensitivity analysis** of these parameters. This omission makes it difficult to assess the robustness and generalizability of the proposed method across different evaluation tasks or model architectures.

4. The reason why reinforcement learning is used instead of SFT as the training paradigm in this article is not clear. The innovation of the paper is incremental. As far as I know, this is not the first role-playing dataset of speech modalities.

Please compare and cite the following articles:
> [1] Character-LLM: A Trainable Agent for Role-Playing. EMNLP 2023

> [2] Neeko: Leveraging Dynamic LoRA for Efficient Multi-Character Role-Playing Agent. EMNLP 2024

> [3] A Systematic Framework for Evaluating and Enhancing the Plot-Progression Capabilities of Role-Playing Agents. ACL2025

> [4] OmniCharacter: Towards Immersive Role-Playing Agents with Seamless Speech-Language Personality Interaction. ACL2025

**Questions:**

1.  **Defining the "Absolute Quality" of Standard Samples:** The Standard Alignment RL mechanism relies on the average reward of high-score standard samples ($r_u$). How is the **"absolute quality"** of these standard samples reliably ensured? If the highest-scoring samples are only of moderate quality due to data limitations, using them as an absolute standard could still lead to inaccurate reward scaling. Can the authors elaborate on the selection criteria beyond simply having the "highest score" to guarantee the standard’s high fidelity?
2.  **Expert Model Weight Merging Strategy:** The paper states that expert model weights are merged "according to the relevance and frequency of the evaluation tasks." Please detail whether this **weight assignment is empirical, manually set, or determined by a data-driven optimization method.** If the weights are subjectively set, please explain how this manual intervention avoids introducing unintended biases into the overall evaluation fairness.

3. What are the algorithmic differences or innovations in the MOE or weight merging mechanism adopted in this paper compared to the approaches in "Mixture-of-personas language models for population simulation" and "Leveraging Dynamic LoRA for Efficient Multi-Character Role-Playing Agent"?

---

### Official Review · Reviewer_nBbw · 2025-10-31

**Soundness:** 3
**Presentation:** 2
**Contribution:** 3
**Rating:** 6
**Confidence:** 3

**Summary:**

This paper addresses the significant challenge of evaluating speech-based role-playing agents (RPAs), where fidelity depends on both semantic content and vocal characteristics. The authors make two main contributions:
1.  **RoleChat:** A new, large-scale (14,032 samples, 50 characters) evaluation dataset for voice-based RPAs. This dataset is annotated with detailed reasoning across five distinct dimensions: Logical Coherence, Content Relevance, Context Consistency, Emotional Appropriateness, and Style Alignment.
2.  **RoleJudge:** A novel evaluation framework based on the Qwen2-Audio model. Its core technical novelty is a multi-stage training paradigm featuring **Reinforcement Learning with Standard Alignment**. This method trains separate "expert models" for each of the five dimensions and introduces a "standard reward" ($r_u$) derived from high-quality samples to scale the advantage function. This is designed to mitigate reward misalignment and prevent the model from optimizing for the "best-of-the-worst" in a bad batch. The expert models are finally merged via weight averaging.

Experiments demonstrate that RoleJudge significantly outperforms strong baselines, including proprietary models like Gemini 2.5 Pro and GPT-40-audio, on the RoleChat test set, particularly in the crucial dimensions of Content Relevance and Style Alignment. The results are further supported by human A/B tests.

**Strengths:**

The paper's primary strength is that it provides a complete and well-executed solution to a timely and difficult problem. It correctly identifies the limitations of existing text-only and uni-dimensional audio evaluators for the complex task of RPA assessment. The **RoleChat dataset** is a significant, high-quality contribution that will likely benefit the community. The **RoleJudge** framework is technically sound, and its core novelty—**Reinforcement Learning with Standard Alignment**—is an intelligent method to improve reward signal quality and avoid local optima, a common pitfall in training LLM-as-judge models. The paper is extremely clear and the empirical results are strong, with thorough ablations and comparisons against top-tier proprietary models.

**Weaknesses:**

1.  **Evaluation Circularity and Dataset Bias:** The most significant concern is the potential for dataset-specific bias. The RoleChat dataset was constructed using a pipeline that involves LLMs (GPT-4) for generating reasoning and scores. The RoleJudge model is then trained on this data. The primary evaluation (Table 1) shows that RoleJudge is good at predicting the labels *from its own dataset*. This is a form of circularity. It's possible that RoleJudge has simply become very good at mimicking the specific biases and patterns of the GPT-4 annotator, rather than learning a "true" or "general" sense of character fidelity. The human A/B test is a good step to mitigate this, but it is small-scale (10 volunteers, 100 total samples) and cannot fully validate the model's generalizability. A stronger validation would involve testing RoleJudge's zero-shot performance on a completely different, independently created dataset of human preferences for RPA speech.

2.  **Rationale for "Merge-of-Experts" vs. Multi-Task Learning:** The paper's method involves training five separate expert models and then merging their weights via averaging. This is computationally expensive, requiring five full RL training runs. The ablation study (Table 3) includes a baseline "without Weight Merging" which is described as "training a single model on all data". This baseline performs significantly worse (78.81 Overall ACC) than even the SFT-only merged model (82.10 Overall ACC, inferred from the top row). This implies that a naive multi-task learning approach (training one model to predict all 5 scores) suffers from strong negative interference. This is an interesting finding, but the paper does not explore *why*. Why is simple weight merging so much better than joint training? Is it task conflict? The paper adopts the merging approach as a solution but doesn't deeply analyze the problem it's solving.

3.  **Hyperparameter Sensitivity:** The core "Standard Alignment" method introduces several new hyperparameters, particularly for the scaling function in Eq 2 ($a, b, \alpha$) and the reward balancing in Eq 3 ($\pi$). The appendix lists the values used ($a=0.5, b=1.5, \alpha=8, \pi=0.8$)[cite: 1049], but there is no sensitivity analysis or justification for these choices. The method's performance could be highly dependent on this specific tuning, and it would strengthen the paper to show how robust the approach is to these new hyperparameters.

**Questions:**

1.  **On Dataset Bias (W1):** Given that the RoleChat labels were guided by GPT-4, how can you be sure that RoleJudge's high accuracy on the test set isn't primarily reflecting its ability to mimic the GPT-4 annotator's biases? The A/B test  is a good, but small, step. Have you considered any cross-dataset validation to demonstrate RoleJudge's generalization to human preferences *not* captured in your dataset?

2.  **On Weight Merging (W2):** The ablation in Table 3 suggests that joint multi-task training (the "w/o Weight Merging" row) performs very poorly compared to merging SFT-only models. This implies strong task interference. Could you elaborate on this? Why do you believe joint training fails here, and why is simple weight averaging a better solution than other multi-task optimization strategies (e.g., gradient balancing, etc.)?

3.  **On Standard Alignment Hyperparameters (W3):** Your Standard Alignment method introduces four new hyperparameters ($a, b, \alpha, \pi$). Could you provide a brief sensitivity analysis or intuition for these values? For instance, how does the model's performance change with different values of $\alpha$ (the sharpness of the sigmoid) or $\pi$ (the reward balancing)?

4.  **On Standard Sample Selection:** In the RL phase, you use an average reward $r_u$ from "standard samples" as a scaling factor. How are these standard samples defined and selected during training? Is it a fixed set of high-quality data, or is it the single "standard sample" identified for each query during dataset construction?

5.  **On "Authentic" vs. "Synthetic" Data:** The RoleChat dataset contains both "authentic" speech from audiovisual works and "synthetic" data generated by LLMs and TTS. Does RoleJudge perform differently on these two subsets in the test set? I would be interested to know if its accuracy is higher on the synthetic data (which is more "in-domain" with the generation process) or if it generalizes equally well to the authentic samples.

---

### Official Review · Reviewer_iwk1 · 2025-10-31

**Soundness:** 1
**Presentation:** 2
**Contribution:** 1
**Rating:** 0
**Confidence:** 3

**Summary:**

This paper describes fine-tuning Qwen2-Audio-7B-Instruct on top of Gemini-2.5 Pro and GPT-4 outputs for evaluating audio role-playing models. The outputs include CoT and final scores (with an unspecified scale). There are 5 evaluation dimensions in different modalities. The resulting synthetic outputs are collected as the RoleChat dataset after some undisclosed manual filtering procedure ("multiple volunteers were recruited to manually verify and refine the results").

For each evaluation dimension, a separate model is fine-tuned. The fine-tuning has 2 steps: SFT and GRPO. Rewards for GRPO are score and format accuracy. GRPO also has a tweak: authors scale relative advantages using a reward computed on reference samples for the same query.

After fine-tuning, all models are merged into one.

Experiments on the RoleChat test set show that this merged model is outperforming open‑ and closed‑source baselines and winning human A/B tests. Ablations suggest benefits from RL and the standard alignment trick.

**Strengths:**

The task makes sense. There is a clear problem formulation of multidimensional role‑playing speech evaluation. High‑level pipeline and figures are helpful. There is some originality (the "standard alignment" GRPO trick).

**Weaknesses:**

Major points:

1. Zero reproducibility: prompts, sampling settings, seed variance, and code/data release plans are missing. There are no supplementary materials either.

2. Everything related to scores is not specified. What was the scale of scores? Figure 1 suggests it is from 1 to 5, but it is never explicitly specified. If it is from 1 to 5, why exactly?

3. Why do you calculate accuracy for your scale? The difference between 1 and 5 is much more important than between 3 and 4.

4. Everything related to human annotators is not specified. What was the filtering procedure? How many samples were removed? How many people were filtering outputs? Was there an overlap? How did you recruit volunteers? Do you pay them? Who are those people? As for the volunteers for A/B testing, are they the same people? And so on, the whole methodology section is just missing.

5. GRPO usage is not justified enough. It might be that SFT hyperparameters are just underexplored, batch size of 4 might be a reason too. The "generalization" in question for which RL might be required is just new characters. Proper SFT should handle new characters pretty well.

6. No statistical uncertainty (no CIs or tests).

7. ASR dependency is under‑specified: text baselines use ASR inputs, but the ASR system, WER on test, and its effect on each dimension are not reported; this may unfairly handicap text‑only baselines.

8. "Standard alignment" is intuitive but under‑analyzed. No ablations on scaling hyperparameters (a,b,α,π), group size, or standard‑sample selection quality. And overall, there are no ablations for hyperparameters even in the SFT case.


Minor points:

1. Equation 1 (the one with the SFT loss) is just wrong. You don't predict the whole output conditioning on the whole input; it is oversimplified. Why is it even in the paper? Do you really think the readers have never seen the cross-entropy loss?

2. "MULTIDIMENSIONAL EVALUATION FRAMEWORK" in caps in Appendix B.

3. Why are format differences a thing? Just use structured outputs (xgrammar, for instance).

4. Weight‑merging strategy is not detailed (layer‑wise? scaling? per‑module weights) and lacks a clean ablation isolating its incremental gain (Table 3 does not compare with/without merging under otherwise identical settings, the readers have to compare with Table 1).

**Questions:**

All the questions are in the "Weaknesses".

Additional questions:

1. A/B test protocol: Did raters listen to the audio samples in a blind setup, or did they look at model scores?

2. Given the use of copyrighted character audio and zero‑shot voice cloning, what licenses/permissions were obtained? Will audio be released, and under what license? If not, can you release transcripts, prosody features, or synthetic surrogates?

3. How are "volunteers" compensated?

**Details Of Ethics Concerns:**

The paper does not adequately discuss legal/ethical issues around using copyrighted dialogue and cloned voices from film/TV characters, nor consent or licensing for redistribution.

The paper mentions "inviting volunteers" for data annotation, and does not mention any details of how these "volunteers" are hired and compensated.

---

### Official Review · Reviewer_Fw2p · 2025-11-01

**Soundness:** 2
**Presentation:** 2
**Contribution:** 3
**Rating:** 4
**Confidence:** 5

**Summary:**

This paper addresses the challenge of evaluating the performance of Audio Role-Playing Agents (RPAs). Traditional text-based evaluation methods fail to capture the rich non-semantic information inherent in speech—such as emotion, intonation, and style—which is crucial for effective character portrayal.

To this end, the authors propose two core contributions:

1. RoleChat Dataset: The first large-scale dataset specifically designed for evaluating audio role-playing conversations. It contains detailed profiles for 50 distinct characters, multi-turn dialogue histories, and audio responses generated by various models. Each sample is meticulously annotated across five dimensions (logical coherence, content relevance, contextual consistency, emotional appropriateness, stylistic alignment) with detailed reasoning processes.

2. RoleJudge Evaluation Framework: A multidimensional evaluation framework based on audio large language models. It employs a multi-stage training strategy (SFT+RL), ultimately converging into a unified evaluation model. Its core innovation introduces high-quality “ground-truth samples” as absolute reward benchmarks within reinforcement learning. This addresses the inefficiency of pure relative preference learning when all candidate responses exhibit low quality.

Experimental results demonstrate that RoleJudge significantly outperforms multiple existing baseline models (including open-source and proprietary text and multimodal models) across multiple evaluation dimensions. It also exhibits higher consistency with human preferences in A/B human evaluations, validating the framework's effectiveness.

**Strengths:**

- The construction of the RoleChat dataset not only provides multi-dimensional scoring metrics but also incorporates CoT-style reasoning processes, offering valuable resources for training more interpretable evaluation models. The dataset includes both authentic speech and diverse synthetic speech samples, providing models with abundant positive and negative examples—a critical factor for training a robust evaluator.
- The paper's proposed “Standard Alignment” mechanism represents a significant improvement over existing preference-based reinforcement learning approaches. By introducing an absolute “standard reward” to scale relative advantages, this method enables more stable optimization and prevents models from getting stuck in local optima when high-quality feedback is scarce. This approach is conceptually clear, cleverly implemented, and demonstrates strong generalizability.

**Weaknesses:**

- Section 3.4 mentions that the five expert models are integrated using a weighted parameter merging strategy, with weights assigned “based on the relevance and frequency of evaluation tasks across each dimension.” This is a critical step, yet the description remains overly vague. As a result, readers cannot discern how the weights are specifically quantified—whether they are manually set hyperparameters or learned through some algorithm—which undermines the reproducibility of the method to some extent.
- Training five independent expert models and then merging them sounds computationally intensive. The paper does not discuss the computational resource differences between this approach and training a unified multi-task evaluation model. For an evaluation framework, inference efficiency and training cost are important considerations; such a discussion would help provide a more comprehensive assessment of the method's practicality.
- Despite its rigorous construction process, the generation and annotation of the RoleChat dataset rely heavily on existing large language models such as GPT-4 and Gemini 2.5 Pro. This may transmit inherent biases from these “teacher models”—such as stylistic preferences and value systems—into the dataset, thereby compromising the objectivity and fairness of RoleJudge evaluations.

**Questions:**

- Instead of training five separate expert models, have you considered directly training a single, end-to-end multi-task model to predict scores across all five dimensions simultaneously? How do you assess the performance and training cost advantages or disadvantages of the “expert models + fusion” approach compared to the end-to-end method?
- The construction process of RoleChat heavily relies on GPT-4 and Gemini. Have you evaluated potential bias issues arising from this dependency? For instance, might RoleJudge favor responses stylistically closer to its “teacher model”? Are there future plans to introduce more diverse human annotations to mitigate this problem?
- In equation (2), the scale(ru) function introduces hyperparameters such as a, b, and α. How sensitive is your method to the settings of these hyperparameters?

---

### Note · Authors · 2026-01-08

I have read and agree with the venue's withdrawal policy on behalf of myself and my co-authors.